# Comparison of Surgical Interventions for Endometrioma: A Systematic Review of Their Efficacy in Addressing Infertility

**DOI:** 10.3390/biomedicines12122930

**Published:** 2024-12-23

**Authors:** Alexandra Ioannidou, Nikolaos Machairiotis, Sofoklis Stavros, Anastasios Potiris, Theodoros Karampitsakos, Athanasios G. Pantelis, Petros Drakakis

**Affiliations:** 1Third Department of Obstetrics and Gynecology, Attikon Hospital, Medical School, National and Kapodistrian University of Athens, 1 Rimini, 124 62 Athens, Greece; alexandra.ioannidou97@gmail.com (A.I.); sfstavrou@yahoo.com (S.S.); apotiris@gmail.com (A.P.); theokarampitsakos@hotmail.com (T.K.); pdrakakis@hotmail.com (P.D.); 2Surgical Department of Obesity & Metabolic Disorders, Psychiko Clinic, Athens Medical Group, Andersen Str., 1, Psychiko, 115 25 Athens, Greece; ath.pantelis@gmail.com

**Keywords:** endometriosis, endometrioma, anti-mullerian hormone, laparoscopic cystectomy, CO_2_ laser vaporization, plasma energy ablation, sclerotherapy, GnRH agonist treatment, assisted reproduction treatment

## Abstract

**Background**: Endometriosis is characterized by the presence of endometrial tissue outside the uterus. Beyond medical treatment, surgical intervention is also a viable consideration. However, current guidelines do not clearly indicate whether laparoscopic cystectomy, ablative methods (CO_2_ laser vaporization, plasma energy), or sclerotherapy is the preferred option. **Methods**: We conducted searches in two databases (PubMed and Europe PMC) to retrieve articles containing the keywords ‘surgical intervention for Endometrioma, ovarian reserve, pregnancy rates, fertility’, published between 1 January 2000 and 31 December 2023. We included articles presenting information on surgical intervention for endometrioma and its correlation with infertility parameters. Articles describing conservative treatment were excluded. Data were extracted by two authors using predefined criteria. **Results**: The initial database search produced 1376 articles, which were narrowed down to 41 relevant articles meeting the eligibility criteria. **Conclusions**: Laparoscopic cystectomy appears to impact postoperative anti-mullerian hormone levels, showing a stronger correlation with larger cysts and individual factors. CO_2_ laser vaporization demonstrates favorable results compared to traditional cystectomy. Combining GnRH agonist treatment with assisted reproduction treatment after cystectomy could be considered an alternative method. Plasma energy causes less damage to ovarian function, with pregnancy outcomes comparable to cystectomy. Sclerotherapy shows promising results for ovarian reserve preservation, recurrence rates, and safety. Further studies comparing these techniques are necessary to provide guidance to clinicians.

## 1. Introduction

Endometriosis manifests through the growth of endometrial tissue outside the uterus, exhibiting in diverse forms such as peritoneal lesions, ovarian cysts (endometrioma), deep-seated variants, and extrapelvic involvement, among others.

Among the intriguing historical theories in medicine, the retrograde menstruation hypothesis stood as a prominent explanation. It proposed that fragments of the endometrial tissue traveled backward through the fallopian tubes, settling in the pelvis and adhering to the peritoneum and abdominal organs. These implants were believed to thrive, particularly under estrogens’ influence, leading to persistent inflammation and the formation of adhesions. However, contemporary research is steering away from the retrograde menstruation theory. Newer perspectives are emerging, proposing a genetic and epigenetic origin associated with intracellular aromatase activity. This updated line of thinking seeks to understand the role of genetics and how gene expression and cellular activity, particularly aromatase, might play a pivotal role in the development of related conditions. As science progresses, these evolving theories shed new light on the complexities of these medical phenomena [1].

Its impact extends far beyond the realms of pain and disrupts everyday life, casting a shadow of concern—its intricate link with infertility [2]. Despite affecting 6–10% of women in their reproductive years, a staggering 50% of those seeking fertility treatment harbor endometriosis lesions [3].

Endometriomas stand out as the most prevalent form of endometriosis manifestation. Yet, lesions can extend beyond the ovaries, appearing throughout the abdominal region. They might be found on the bowel, within previous surgical scars, and in rare instances, even in distant sites like the cerebellum [4]. Statistics suggest that approximately 17 to 44% of women diagnosed with endometriosis will encounter an endometrioma [5].

These lesions, colloquially known as “chocolate cysts”, owe their name to the thick, dark brown fluid they contain. However, beyond their appearance, endometriomas serve as indicators of a more severe state of the disease in individuals with endometriosis [5]. Research on endometrioma’s effect on the ovarian reserve, specifically concerning serum anti-Müllerian hormone (AMH) levels, is riddled with contradictory findings. Some studies indicate a decrease in AMH levels among women with endometriomas, while others fail to establish significant differences when compared to control groups. To arrive at a conclusive understanding, more extensive and prolonged investigations are imperative [6].

Surgical interventions, notably excision of the cyst’s capsule (cystectomy), exhibit a potential to enhance postoperative pregnancy rates compared to drainage and ablative methods [7]. Yet, the delicate ovarian tissue and follicles face possible harm from surgical excision. Fears of ovarian failure after cystectomy have prompted healthcare providers to embrace ablative approaches. In this surgical method, rather than entirely removing it, the “pseudo-capsule” undergoes ablation using energy sources with limited thermal dispersion. A new approach, the ‘one step’ or ‘three step’ laser vaporization, employing CO_2_ laser technology, emerges as a promising option by minimizing energy usage and consequently decreasing potential damage. It stands as a ray of hope in preserving fertility while effectively managing endometrioma [8,9]. Another alternative is sclerotherapy. The purpose of sclerotherapy is to eliminate the pseudo-capsule of an endometrioma by introducing alcoholic substances into it. This method is regarded as a cost-effective and safe way to protect healthy ovarian tissue. The procedure involves injecting a sclerosing substance directly into the cyst, followed by rinsing or retaining it. Hydrosoluble dehydrating antiseptic fluids have the capability to alter the proteins in microorganism envelopes and dissolve the lipids of their capsules. Ethanol is often preferred over other sclerosing substances due to its superior track record in managing renal and hepatic cysts [10].

In the latest ESHRE (European Society of Human Reproduction and Embryology) guidelines for endometriosis there were no randomized controlled trials (RCTs) comparing fertility outcomes post-surgery for endometrioma versus expectant management, and studies on surgery indication based on cyst size were lacking [11]. This, combined with the lack of specific guidelines on the subject, makes the need for research that deals with the comparison of these surgical techniques imperative. This review aims to investigate the possible impact of available surgical interventions for endometrioma at fertility parameters and outcomes.

## 2. Methods

The conduct of this systematic review adhered to the guidelines set by the Preferred Reporting Items for Reviews and Meta-Analyses (PRISMA) Statement. The protocol was submitted via protocols.io (DOI: dx.doi.org/10.17504/protocols.io.kqdg3227pv25/v1, accessed on 16 October 2024) and was accepted and published before the final search.

### 2.1. Data Sources and Search Strategy

A search was conducted across two databases—PubMed and Europe PMC—to collect articles containing the keywords ‘surgical intervention for endometrioma, ovarian reserve, pregnancy rates, fertility’ published between 1 January 2000 and December 2023.

### 2.2. Eligibility Criteria for Articles of Inclusion

A total of 1377 articles were retrieved from the search of the databases. They were screened by two reviewers (A.I., N.M.), from which 767 articles were excluded to identify 610 articles of interest. Of these 610, a further 561 were excluded based on the eligibility criteria. The articles were limited to those that were written in English and only full research articles. Abstracts presented in scientific meetings were excluded as well as review articles. Only articles presenting information about surgical intervention for endometrioma and their correlation with infertility parameters were included. Any articles describing conservative treatment for endometrioma were also excluded by screening titles and abstracts for relevant keywords. To avoid bias, articles that referred to the combined treatment of endometrioma were considered. This review is based on 41 remaining articles that fulfilled the eligibility criteria, as shown in Figure 1.

### 2.3. Data Extraction

Specific data were extracted from each publication, including publication date, authorship, studied population, methodologies employed, criteria for inclusion or exclusion (I/E), sample type, and primary outcome measures.

## 3. Results

The studies reviewed, along with their corresponding outcomes, are summarized in Table 1.

### 3.1. Laparoscopic Cystectomy

In a prospective cohort study involving 100 women who underwent a laparoscopic ovarian cystectomy for either endometriomas or nonendometriotic benign ovarian cysts, both cohorts exhibited a reduction in anti-Müllerian hormone (AMH) levels following surgery. The preoperative AMH levels (4.97 ± 2.83 ng/mL, mean ± SD) decreased to 3.33 ± 2.08 ng/mL postoperatively, indicating a comparable impact on the ovarian reserve regardless of cyst type (*p* = 0.36). The correlation between bilaterality, specifically bilateral cystectomy, and the decline in postoperative AMH levels was significant, particularly in cases of endometriomas. While the AMH decline was similar between unilateral endometrioma and benign cyst groups, it was notably higher in cases of bilateral endometriomas (49.65 ± 4.84 vs. 26.69 ± 3.31, *p* < 0.001) [32].

Another prospective cohort study sought to explore the repercussions of surgical intervention (specifically laparoscopic cystectomy) for endometriomas on the ovarian reserve. The investigation encompassed 54 women of reproductive age diagnosed with unilateral or bilateral endometriomas. AMH levels were assessed before the surgery, as well as at 6 and 12 months postoperative. The key findings illuminated a reduction in serum AMH levels post-surgery, particularly evident in cases of bilateral endometriomas (baseline for unilateral 3.31 ± 1.74, 6 months 1.43 ± 1.01, 12 months 1.72 ± 1.23, overall *p*-value < 0.001/baseline for bilateral 2.55 ± 1.87, 6 months 0.98 ± 0.91, 12 months 0.89 ± 0.82, overall *p*-value < 0.001). Age, baseline AMH levels, and the presence of bilateral endometriomas emerged as significant predictors of post-surgery AMH levels. Younger patients tended to exhibit higher AMH levels post-surgery, and baseline AMH levels consistently predicted postoperative AMH levels. Notably, patients with unilateral endometriomas tended to retain higher postoperative AMH levels. The study also delved into pregnancy outcomes, revealing a noteworthy number of women achieving spontaneous pregnancies post-surgery despite the observed decline in the ovarian reserve. This suggests a quantitative impact on the ovarian reserve rather than a qualitative one [39].

In another piece of research, 85 women experiencing subfertility and who underwent their initial laparoscopic endometriotic cystectomy were investigated. The findings indicated no statistically significant difference in the ovarian reserve when comparing pre- and postoperative biochemical parameters (mean FSH: 7.24 ± 1.21 mIU/mL, 6.37 ± 1.8 mIU/mL, and mean LH: 7.23 ± 1.51 mIU/mL, 6.6 ± 2.3 mIU/mL, respectively, *p* = 0.29). Similar results were observed for ultrasound parameters [mean residual ovarian volume of the operated ovary [cm^3^] 8.5 ± 5.3 and 7.4 ± 5.8, respectively, *p* = 0.19/mean stromal PSV (peak systolic velocity) of the operated ovary 6.8 ± 4.57 and 7.1 ± 3.55 cm/s, respectively, *p* = 0.488]. The sole parameter exhibiting a significant difference was the mean AFC (pre- and postoperatively 3.3 ± 1.9 and 4.1 ± 1.5, respectively, *p* = 0.001). Histopathological analysis revealed follicle loss in a segment of operated cyst walls, particularly in smaller cysts (for cysts < 5 cm 40% experienced follicle loss, while for cysts ≥ 5.1 18.86% exhibited follicle loss, *p* = 0.03) [33].

During a clinical trial, women under 40 experiencing infertility with asymptomatic endometriomal cysts (2 to 6 cm in diameter) and undergoing intracytoplasmic sperm injection (ICSI) were recruited. The participants were randomly allocated into two groups: one group underwent a laparoscopic cystectomy along with GnRH-agonist treatment, while the other group received GnRH-agonist treatment alone. The surgical procedure involved cyst wall resection followed by three doses of Diphereline (GnRH-agonist). Both groups underwent ovarian stimulation for ICSI. Results indicated higher rates of chemical pregnancies (combined group 48.48%, GnRH-a alone 30.8%, *p* = 0.124) and clinical pregnancies (combined group 36.36%, GnRH-a alone: 25.6%, *p* = 0.325), as well as live births in the combined treatment group (combined group: 36.36%, GnRH-a alone: 23.1%, *p* = 0.292), but without statistically significant differences (combined/GnRH-a alone: adjusted OR for live birth rate 1.08, *p* = 0.961). Ultimately, the study suggests that for infertile women with endometriomal cysts, combining laparoscopic cystectomy and GnRH-agonist treatment before assisted reproductive technology (ART) may enhance pregnancy rates, although larger studies are necessary to validate these findings [44].

A study conducted at the “Papageorgiou” University Hospital in Greece aimed to compare the effects of two laparoscopic approaches—one-step cystectomy and three-step CO_2_ laser vaporization—on the ovarian reserve in the treatment of ovarian endometriomas. The results indicated a significant (*p* = 0.026) reduction in AMH levels with cyst stripping (from 3.9 to 2.9 ng/mL) compared to CO_2_ laser vaporization (4.5 to 3.9 ng/mL, *p* not significant). AFC exhibited a notable increase in the CO_2_ laser vaporization group (from 1.27 to 4.36, *p* = 0.002). Parameters such as ovarian volume, hormone levels, and ultrasound findings did not show significant differences between the two groups post-treatment. The study demonstrated that ovarian cystectomy led to a diminished ovarian reserve in reproductive-age women, whereas CO2 laser vaporization exhibited enhanced follicular recruitment in the treated ovary. Both AMH and AFC proved to be sensitive indicators for evaluating the impact of these surgical techniques on the ovarian reserve [25].

In another study, individuals aged 18 to 45 underwent a laparoscopic cystectomy and were randomly assigned to three groups employing distinct hemostasis techniques. Analysis of the three groups uncovered noteworthy differences (*p* < 0.05) in ovarian reserve markers, specifically AFC and PSV. This suggests that the hemostasis technique involving sutures yielded superior outcomes in preserving ovarian function compared to bipolar electrocoagulation and ultrasound scalpel [36].

In a prospective study carried out in a Beijing hospital, findings suggested that surgical intervention might lead to a reduction in the ovarian reserve. This inference was drawn from the observation that the AMH levels were comparatively lower in the surgery group (1.96 ± 1.00 ng/mL) than in the non-surgery group (3.71 ± 2.21 ng/mL, *p* = 0.01). Conversely, the impact of surgery did not exhibit a statistically significant effect on the success rates of in vitro fertilization (IVF), with *p*-values indicating non-significance across all parameters [42].

In a prospective cohort study involving 65 patients who underwent a laparoscopic excision of ovarian endometriomas, the results revealed a notable decline in the ovarian reserve, particularly evident in the levels of serum AMH and FSH six months post-surgery (*p* < 0.001). The reduction in AMH levels were more conspicuous in cases where endometriomas were bilateral (67% for bilateral and 57% for unilateral, *p* = 0.039) or exceeded 5 cm in size (65.7% for >5 cm and 41.3% for <5 cm, *p* = 0.003) [28].

The findings of another study revealed a substantial decrease in serum AMH levels among patients with endometrioma compared to those without it (*p* < 0.001). Furthermore, prior cystectomy accentuated this reduction in AMH levels (*p* < 0.001). In cases of bilateral endometriomas and cystectomies, the decline in serum AMH levels was more pronounced than in unilateral instances (1.01 ± 0.11 vs. 1.48 ± 0.14 ng/mL, *p* < 0.05). Notably, AMH levels exhibited a significant decrease three months post-cystectomy when compared to preoperative levels, underscoring the impact of surgery on the ovarian reserve (from 3.95 ± 0.42 ng/mL to 2.01 ± 0.21 ng/mL 3 month postoperative, *p* < 0.01) [15].

Other studies demonstrated a significantly reduced yield of recovered oocytes from ovaries that underwent endometrioma excision (2.2 ± 2.0) compared to their healthy counterparts (5.1 ± 3.3, *p* = 0.009). These findings support previous observations, highlighting a quantitative impact of the excision procedure. However, no noteworthy differences were observed in normal fertilization rates (63.6% vs. 69.5%), top-quality embryos (40.0% vs. 49.0%), clinical pregnancy rates (40.0% vs. 25.0%), or ongoing pregnancy rates (20.0% vs. 20.8%) between oocytes retrieved from operated and healthy ovaries. These results suggest that the surgery did not exert a qualitative effect on the developmental competence of the recovered oocytes [19].

Another study centered on laparoscopic endometrioma cystectomy and its impact on the ovarian reserve, specifically assessed through serum AMH levels, revealed notable changes in preoperative AMH levels following surgery. A significant decrease was observed [from 3.77 ng/mL to 1.60 ng/mL (*p* < 0.001), 1.66 ng/mL (*p* < 0.001), 1.67 ng/mL (*p* < 0.001), and 1.72 ng/mL (*p* < 0.001) at 1, 3, 6, and 12 months postoperatively, respectively], but a recovery was evident at 12 months (*p* < 0.05), particularly in cases of unilateral endometriomas. In contrast, bilateral endometriomas exhibited a sustained decline in AMH levels, even at 12 months post-surgery (with a rate of decrease of 48.0%, 48.2%, 43.4%, and 34.9% in unilateral endometriomas and 78.0%, 85.0%, 83.6%, and 83.2% in bilateral endometriomas at 1, 3, 6, and 12 months postoperative, respectively). Notably, approximately 29.2% of infertile patients reported spontaneous pregnancies within 12 months post-surgery, indicating an enhanced likelihood of conception following the procedure [48].

A research study carried out at Kasr El Aini Hospital, Cairo University, spanning from October 2016 to January 2019, focused on women aged 20 to 35 who were undergoing laparoscopic treatment for ovarian endometriomas. The participants were categorized into four groups, each subjected to different procedures: drainage, cystectomy, drainage with Surgicel insertion, or cystectomy with Surgicel insertion. Following the treatments, all four groups exhibited a reduction in AMH levels, with Group DS displaying the most significant decrease compared to the other three groups. Additionally, the AFC in the operated ovary showed a notably higher decrease in Group D in comparison to Group CS (*p* = 0.021) [43].

A total of 57 infertile individuals with a history of cystectomy were included in a study, and their reactions to COH (controlled ovarian hyperstimulation) were contrasted with those of a control group consisting of 99 patients with infertility due to tubal factors. The findings revealed that although the group that underwent unilateral or bilateral cystectomy demonstrated results comparable to the control group in terms of COH response and pregnancy rates, the bilateral cystectomy subgroup necessitated a higher dosage of FSH and produced fewer oocytes (*p* < 0.05) [12].

Other research investigated how the removal of ovarian endometriomas through laparoscopic excision affects subsequent in vitro fertilization (IVF) treatment in 85 women, categorized based on the size of the excised cysts (<4 cm or ≥4 cm). In both groups, a reduction in the operated ovaries was observed, but this decrease was notably more significant in individuals with cysts ≥ 4 cm. Statistical analysis confirmed these distinctions (AFC 2.5 ± 3.9 vs. 0.7 ± 2.7, *p* = 0.011; No. of dominant follicles 1.9 ± 2.9 vs. 0.8 ± 2.7, *p* = 0.039; no. of oocytes retrieved 3.3 ± 3.6 vs. 0.9 ± 4.9, *p* = 0.006). Larger cyst sizes were associated with a higher likelihood of a reduced number of oocytes retrieved (*p* < 0.05). The findings of the study imply a potential connection between ovarian damage after endometrioma excision and the size of the cyst [17].

A study aimed to compare AMH levels between three-step surgery with dienogest and one-step surgery without additional meds, following patients for a year post-surgery. Three-step surgery with dienogest showed no statistical difference in AMH decline at 9–12 months post-surgery, but one-step did (*p* = 0.01 and *p* = 0.16 for one-step and three-step group, respectively). Peritoneal fluid analysis revealed varying cytokine changes after dienogest, suggesting individual responses [45].

Another research study involved the inclusion of 25 women diagnosed with unilateral endometriomas. Laparoscopic excisional surgery was carried out with a focus on removing the cyst capsule while minimizing harm to the healthy ovarian tissue. Results indicated a 24% reduction in AMH concentration (*p* < 0.01) and an 11% decline in AFC at 1 month post-operation compared to preoperative levels (*p* = 0.01). These reductions in AMH and AFC persisted for 6 months post-operation [29].

In a prospective investigation carried out at Yas and Arash Hospitals, the objective was to assess the influence of laparoscopic excision of ovarian endometrioma on the ovarian reserve. The study identified a notable reduction in AMH levels after the surgical procedure (*p* < 0.000). Factors that appeared to predict the rate of AMH decline included CA 125 (*p* = 0.016) and the grade of endometriosis (*p* = 0.05) [49].

A cohort study was carried out by pairing women with and without ovarian minor anomalies (OMA) based on their similar ages and serum AMH levels. The study utilized data from the initial IVF/ICSI cycle for each participant. Significantly, prior surgery for OMA substantially heightened the risk of poor ovarian response (POR) during ovarian stimulation, as indicated by univariate analysis (odds ratio (OR) = 2.4; 95% confidence interval (CI): 1.3–4.6; *p* = 0.004) and multivariate analysis (OR: 2.2, 95%, CI: 1.1–4.2; *p* = 0.019) [40].

An additional study encompassed 60 women, divided into two groups: those with endometriomas (bilateral and unilateral cases) and those serving as controls. Women with endometriomas exhibited notably lower AFC and serum AMH levels in comparison to those without cysts (*p* < 0.01, *p* = 0.02, respectively). Following surgery, there was a noticeable trend of reduced serum AMH levels at one month, with a statistically significant difference observed solely for the bilateral group (*p* = 0.05). However, at the six-month mark AMH levels significantly decreased once again, specifically for bilateral endometriomas (*p* = 0.02), while AFC remained unchanged in both postoperative groups [30].

In a prospective study carried out at the Department of Obstetrics and Gynecology of ‘Sapienza’ University in Rome, the focus was on exploring the consequences of recurrent surgical interventions for ovarian endometriomas on the ovarian reserve. The AFC and ovarian volumes displayed a significant reduction in the operated ovary compared to the non-operated one, specifically within the recurrent endometrioma group (*p* = 0.002 and *p* = 0.001, respectively). The findings of the study underscored the importance of exercising caution when contemplating repeated surgeries, as there is a heightened risk of compromising the ovarian reserve with a second surgery compared to the initial one [37].

### 3.2. Ablation

In a prospective study, women aged 18 to 45 with unilateral ovarian endometriomas were included and laparoscopic surgery was conducted. The researchers assessed AMH levels before the surgery, at 3 months, and at 6 months post-surgery. The results indicated a significant initial decline in AMH levels 3 months after PlasmaJet ablation (median AMH level variation, −1 ng/mL, *p* = 0.003), followed by a subsequent gradual increase (median AMH level variation 6 months vs. 3 months, 0.7 ng/mL, *p* = 0.01). However, the AMH levels at the final evaluation after 6 months did not revert to preoperative levels but exhibited a trend of increase with no statistical significance (median AMH level variation 6 months vs. preoperative, −1 ng/mL, *p* = 0.26). Despite the anticipation that PlasmaJet ablation would spare ovarian tissue and preserve ovarian function, the initial decrease in AMH levels prompted inquiries about factors influencing postoperative ovarian AMH production [34].

A pilot study conducted at the VU University Medical Centre in Amsterdam sought to evaluate the effectiveness of plasma energy, concerning recurrence, pregnancy outcomes, and postoperative pain. The study enrolled 21 women who underwent an ovarian endometrioma ablation using plasma energy. The results demonstrated a notable reduction in postoperative pain and a swift recovery, suggesting that plasma energy holds promise as a viable technique. The study also reported a postoperative pregnancy rate of 46.2%, comparable to that of stripping cystectomy. The recurrence rate stood at 9.5%, aligning with rates observed in other surgical approaches [20].

There is other research which is centered on women who underwent treatment for ovarian endometriomas utilizing plasma energy between 2009 and 2012. The overall postoperative pregnancy rate showed promise, reaching 61.4% among women with the intention of conceiving, and a substantial portion of pregnancies occurred spontaneously (64.7%). The findings of the study suggested that ablating ovarian endometriomas using plasma energy might preserve ovarian function, potentially resulting in higher postoperative pregnancy rates compared to cystectomy [35].

### 3.3. Laparoscopic Cystectomy vs. Ablative Methods

A cohort study included 62 individuals who underwent laparoscopic surgery for endometriotic cysts. The patients were categorized into distinct groups based on the specific surgical procedures they received: bilateral cystectomy (BC), bilateral vaporization (BV), unilateral cystectomy (UC), and unilateral vaporization (UV). The findings indicated a notable reduction in serum AMH levels post-surgery across all groups. Notably, the BC group exhibited a more pronounced decline compared to the BV group at various time points (1 month, *p* = 0.04; 6 months, *p* = 0.02; 1 year, *p* = 0.02). Failure to recover AMH levels after surgery was linked to individuals aged over 38 years and with a rASRM score exceeding 80 [41].

In a prospective study, infertile women diagnosed with bilateral endometrioma underwent comprehensive examination. A total of 48 eligible patients, meeting specific criteria, underwent laparoscopic procedures to assess the impact of distinct surgical techniques—coagulation and cystectomy—on the ovarian reserve. The study meticulously evaluated AFC and ovarian volumes both before and after the surgical interventions. Following surgery, notable reductions in both AFC and ovarian volume were evident in cases subjected to both coagulation and cystectomy (AFC; cystectomy: preoperative 5.58 ± 1.13 vs. postoperative 3.67 ± 1.26, *p* = 0.001; cauterization: preoperative 5.42 ± 0.77 vs. postoperative 4.75 ± 0.60, *p* = 0.02. Ovarian volume; cystectomy: 13.03 ± 1.13 vs. 6.27 ± 1.95, *p* = 0.01; cauterization: 13.56 ± 1.5 vs. 9.87 ± 2.01, *p* = 0.01), with cystectomy causing a more pronounced decrease (postcystectomy vs. postcauterization for AFC: *p* = 0.001 and for ovarian volume: *p* = 0.005). Furthermore, in 37 patients who underwent IVF, the results indicated fewer dominant follicles (postcystectomy 4.38 ± 0.95, postcauterization 5.05 ± 0.91, *p* = 0.03) and a reduced number of retrieved oocytes from cystectomized ovaries compared to coagulated ones (3.08 ± 0.79 and 3.86 ± 0.88, respectively, with *p* = 0.01) [26].

Research that took place at two university hospitals focused on patients who had undergone a laparoscopy for endometriomas measuring less than 3 cm. Two distinct groups emerged from the study: Group 1, which underwent a cyst lining dissection and biopsy, and Group 2, which underwent a cyst lining biopsy followed by bipolar coagulation. Notably, the group that underwent a cystectomy demonstrated a markedly higher pregnancy rate after one year in comparison to the other group (59.4% and 23.3%, respectively, with a *p*-value of 0.009) [24].

Some research has centered on the comparison between laparoscopic stripping (cystectomy) and CO_2_ laser vaporization in the management of endometriomas. The primary focus was to evaluate their respective impacts on the ovarian reserve. The findings revealed a noteworthy increase in AFC for the treated ovary (*p* < 0.001) following CO_2_ laser vaporization. This increase was observed from 3.6 ± 1.9 at baseline to 8.6 ± 4.2 at the 3 month follow-up, with a 95% confidence interval of 2.8–7.1. In comparison, cystectomy exhibited a less substantial change, with AFC moving from 4.1 ± 2.2 at baseline to 6.3 ± 3.5 at the 3 month follow-up. These results suggest potential advantages in the laser technique for preserving the ovarian reserve. Furthermore, AMH levels experienced a significant decline in the cystectomy group (*p* = 0.012), dropping from 2.6 ± 1.4 ng/mL at baseline to 1.8 ± 0.8 ng/mL at the 3 month follow-up. Conversely, the laser vaporization group showed a stable trend in AMH levels (*p* = 0.09) over the same period. These outcomes underscore the potential benefits of CO_2_ laser vaporization over cystectomy in maintaining the ovarian reserve [8].

A research investigation took place in a French hospital, focusing on the comparison between two distinct surgical approaches for addressing ovarian endometrioma. One approach entailed cystectomy, which involved the removal of the endometrioma cyst, while the alternative method utilized plasma energy for ablation, aiming to eradicate the inner layer of the cyst. The outcomes indicated that plasma energy ablation resulted in significantly lower reductions in ovarian volume (*p* < 0.001) and AFC (*p* < 0.001) when contrasted with cystectomy [16].

Findings from another study indicate that despite both groups having similar demographic and baseline clinical characteristics, disparities were observed in AMH concentrations after surgery in Group 1 (cystectomy) and Group 2 (ablation with bipolar coagulation). Group 1 exhibited a more pronounced reduction in AMH levels (from 4.25 ng/mL to 3.40 ng/mL) compared to Group 2 (from 4.47 ng/mL to 3.95 ng/mL, *p* = 0.04), suggesting a potentially milder impact on the ovarian reserve in the latter. Furthermore, pregnancy rates were comparable between the two groups (71.05% vs. 73.08%, respectively; *p* > 0.99) [21].

Other research enlisted women diagnosed with bilateral ovarian endometriomas and allocated distinct surgical interventions randomly to each ovary. Notably, at the 6 month post-surgery mark, the group treated with the laser exhibited significantly higher AFC and ovarian volume (OV) compared to the stripping group (*p* = 0.05 and *p* < 0.05, respectively). There were no instances of endometrioma recurrence throughout the study period, and a noteworthy proportion of patients (33.3%) achieved pregnancy, particularly in the ovaries subjected to laser surgery [46].

An investigation, conducted between January 2013 and April 2014, aimed to evaluate the effectiveness of two surgical approaches for managing bilateral endometriomas. During laparoscopy, one side of the endometrioma underwent treatment using a conventional stripping technique, while the other side was subjected to a combined excision/ablation technique. The study revealed no significant disparity in recurrence rates between the two techniques at the 6 month mark. Both approaches demonstrated comparable impacts on the ovarian reserve, with only a slight variation in ovarian volume noted after 6 months (*p* = 0.04), indicating a lower volume in the combined technique compared to the stripping technique [38].

### 3.4. Sclerotherapy

In a retrospective cohort study, findings revealed that women with moderate to severe endometriosis who underwent ethanol sclerotherapy (EST) experienced a significantly higher cumulative live birth rate (CLBR) compared to those who did not undergo EST (31.3% vs. 14.5%, *p* = 0.03). Multivariate analysis further confirmed increased odds of live birth in the EST group, with an adjusted Odds Ratio (OR) of 2.68 (95% CI: 1.13–6.36, *p* = 0.02). Additionally, both clinical and biochemical pregnancy rates were markedly higher in the EST group (37.3% vs. 15.9%, *p* = 0.01 and 43.3% vs. 23.2%, *p* = 0.01, respectively). Despite similar characteristics in IVF cycles between the two groups, there was a notable increased CLBR among women who underwent EST [22].

In another retrospective analysis, 108 consecutive patients who had previously undergone surgical intervention for ovarian endometriomas and were seeking treatment for recurrence were included. The findings revealed a notable contrast in recurrence rates between the group treated with ethanol irrigation (32.1%) and the group undergoing ethanol retention (13.3%, *p* < 0.005). Moreover, both groups exhibited an increase in the number of antral follicles, indicating the preservation of ovarian tissue (irrigation group 49.5% and retention group 60.8%, *p* < 0.050) [14].

In a pilot prospective study involving 21 patients, ovarian aspiration followed by ethanol sclerotherapy was conducted before IVF. The findings suggest that the procedure was devoid of complications and deemed safe. Notably, there were no instances of endometrioma recurrence during the IVF induction process. Additionally, the outcomes were successful, as evidenced by the successful retrieval of oocytes and subsequent embryo transfers, resulting in a pregnancy rate of 20% [27].

In a prospective randomized clinical trial comprising 40 patients experiencing recurrent endometriomas, ethanol sclerotherapy was administered to one group, while the other served as a control without undergoing sclerotherapy. The study demonstrated the safety of ethanol sclerotherapy, with a recurrence rate of 20% after a six-month period. Although there were no significant disparities in IVF outcomes between the two groups, the ethanol-treated group exhibited some favorable trends, such as a higher percentage of good-quality embryos, albeit without statistical significance [31].

In an observational study, mean discrepancy of AMH levels before and after laparoscopic sclerotherapy measured at 1.29 ng/mL (*p* < 0.001). A total of 40% of the patients successfully achieved pregnancy. Among the 51 patients who underwent pelvic ultrasound after surgery, 11% experienced a recurrence of endometrioma [23].

### 3.5. Sclerotherapy vs. Laparoscopic

In a separate investigation, the goal was to evaluate and contrast two therapeutic approaches for ovarian endometrioma: sclerotherapy and laparoscopy. Although both treatments resulted in a significant reduction in cyst size, laparoscopy showed a notable decrease in AMH levels after one year (2.48 ± 1.34 vs. 1.62 ± 1.22; *p* < 0.001). Conversely, the sclerotherapy group displayed no significant alterations in AMH levels (2.12 ± 1.05 vs. 2.09 ± 1.01; *p* = 0.120) [50].

The outcomes from a study following surgical resection indicated a reduction in the number of follicles measuring 14 to 17 mm (*p* = 0.003), as well as a decrease in retrieved oocytes (*p* = 0.016), mature oocytes (*p* = 0.010), and fertilized oocytes (*p* = 0.012) when compared to both the AEST group and the control group. However, no noteworthy differences were observed in the counts of follicles exceeding 17 mm on hCG day, the total dosage, and days of gonadotropin utilized for COS, or in various pregnancy rates across the groups [18].

Another study examined the outcomes of women who underwent ethanol sclerotherapy (EST) at ovum pickup and those who chose laparoscopic ovarian cystectomy, monitoring them for a duration of up to seven years. The participants had unilateral endometriomas of 3–6 cm diagnosed by ultrasound. The findings revealed that both groups had similar baseline characteristics, and there were no notable differences in outcomes related to assisted reproductive techniques, encompassing the total dose and duration of gonadotropin therapy, serum estradiol levels, and clinical pregnancy rates. However, the laparoscopic cystectomy group exhibited a slightly higher clinical pregnancy rate and live birth rate, although this difference did not reach statistical significance. In terms of endometrioma recurrence during the follow-up period, the EST group experienced a significantly higher recurrence rate (34.1% in the EST group compared to 14% in the surgery group, *p* = 0.017) [47].

A research study from Japan focused on the management of ovarian endometriomas in infertile patients, exploring various treatment approaches such as cystectomy, aspiration with or without alcohol fixation, and the utilization of assisted reproductive technology (ART). Among patients with bilateral endometriomas who underwent cystectomy (Group B), the cumulative pregnancy rate was 18%, notably lower when compared to a cyst-free control group (44% pregnancies) and patients with unilateral endometriomas and cystectomy, where the rate stood at 43%. In the context of ART, Group B exhibited a reduced number of fertilized oocytes compared to patients with aspiration and bilateral endometriomas (Group b) (*p* < 0.005), along with fewer blastocysts available for transfer (*p* < 0.005). Furthermore, the cumulative pregnancy rate of Group B was lower than that of Group b, although this difference did not reach statistical significance (*p* = 0.052) [13].

## 4. Discussion

The systematic review of studies investigating the impact of surgical interventions for ovarian endometriomas on fertility parameters provides valuable insights into the complex relationship between surgical techniques and reproductive outcomes.

The majority of studies investigating laparoscopic cystectomy highlight a consistent reduction in AMH levels postoperatively [15,25,28,29,30,32,39,42,43,45,48,49], indicating an adverse effect on the ovarian reserve. The decline is particularly pronounced in cases of bilateral endometriomas, suggesting a correlation between cyst laterality and the extent of AMH reduction [15,28,30,32,39,48]. Additionally, factors such as age, baseline AMH levels, and cyst size contribute to the observed variations in postoperative AMH levels [17,28,39]. Despite the decline in the ovarian reserve, some studies suggest favorable pregnancy outcomes, emphasizing the need to consider both quantitative and qualitative aspects of ovarian function [39,48].

Studies comparing different laparoscopic approaches, such as cystectomy versus CO_2_ laser vaporization, highlight significant variations in AMH levels, follicle counts, and ovarian volumes. CO_2_ laser vaporization appears to offer advantages in terms of maintaining the ovarian reserve compared to traditional cystectomy [25]. Furthermore, the choice of hemostasis technique during cystectomy influences the ovarian reserve markers, emphasizing the importance of meticulous surgical considerations [36]. The findings underscore the need for individualized treatment approaches based on patient characteristics and the specific surgical technique employed.

The impact of laparoscopic cystectomy on subsequent ART outcomes, including IVF success rates, chemical pregnancies, and live birth rates, is explored in various studies. Despite a decline in the ovarian reserve, laparoscopic cystectomy does not seem to significantly compromise the overall success of ART procedures [19,42]. Combining laparoscopic cystectomy with gonadotropin-releasing hormone (GnRH) agonist treatment before assisted reproductive technology (ART) may enhance pregnancy rates, providing a potential avenue for improving fertility outcomes in infertile women with endometriomal cysts [44].

Ablative techniques, such as PlasmaJet ablation, present an alternative to cystectomy, demonstrating initial declines in AMH levels followed by a gradual recovery [29]. Studies suggest that plasma energy holds promise for preserving ovarian function and achieving comparable pregnancy rates to traditional cystectomy [8,16,20,21,24,26,37,46]. Furthermore, the reduction in postoperative pain and swift recovery associated with plasma energy ablation highlight its potential as a viable and less invasive therapeutic option [20].

Ethanol sclerotherapy emerges as a potential alternative for managing recurrent endometriomas, demonstrating favorable outcomes in terms of cumulative live birth rates, clinical pregnancies, and antral follicle preservation [14,22,23,27,31]. Sclerotherapy, when compared to laparoscopic cystectomy, shows promising results in terms of ovarian reserve preservation and recurrence rates [18,23,50]. The safety and efficacy of sclerotherapy make it an attractive option for patients seeking fertility preservation while addressing endometrioma recurrence [31].

Comparisons between laparoscopic cystectomy, ablative techniques, and sclerotherapy reveal nuanced differences in their impact on the ovarian reserve, reproductive outcomes, and recurrence rates. Factors such as age, cyst characteristics, and individual patient responses contribute to the variability observed across studies. Additionally, caution is advised when considering repeated surgical interventions, as recurrent surgeries may pose a heightened risk of compromising the ovarian reserve.

Despite the wealth of information provided by these studies, certain limitations exist. Variability in study designs, patient populations, and outcome measures make direct comparisons challenging. Additionally, the follow-up durations across studies vary, making it difficult to assess the long-term implications of surgical interventions on the ovarian reserve and reproductive outcomes.

In brief, the discussion of laparoscopic interventions for ovarian endometriomas underscores the intricate balance between achieving effective cyst management and preserving the ovarian reserve. While laparoscopic cystectomy is commonly associated with a decline in AMH levels, the choice of surgical technique, patient selection, and individual factors contribute to the nuanced outcomes observed across studies. Further research, particularly with longer follow-up periods and standardized outcome measures, is essential for a more comprehensive understanding of the impact of laparoscopic interventions on the ovarian reserve and fertility outcomes in women with endometriomas.

## 5. Conclusions

In conclusion, the systematic review underscores the complex interplay between surgical interventions, ovarian reserve, and reproductive outcomes in patients with ovarian endometriomas. While laparoscopic cystectomy remains a common approach for managing ovarian endometriomas, emerging evidence suggests that alternative methods such as ablative techniques and sclerotherapy may offer promising results with potentially less impact on the ovarian reserve. Bilateral endometriomas have been shown to significantly impact fertility outcomes, often with statistical significance. However, the complex interplay between surgical interventions and fertility outcomes warrants further investigation through larger, well-designed studies to guide clinicians in optimizing patient care.

## Figures and Tables

**Figure 1 biomedicines-12-02930-f001:**
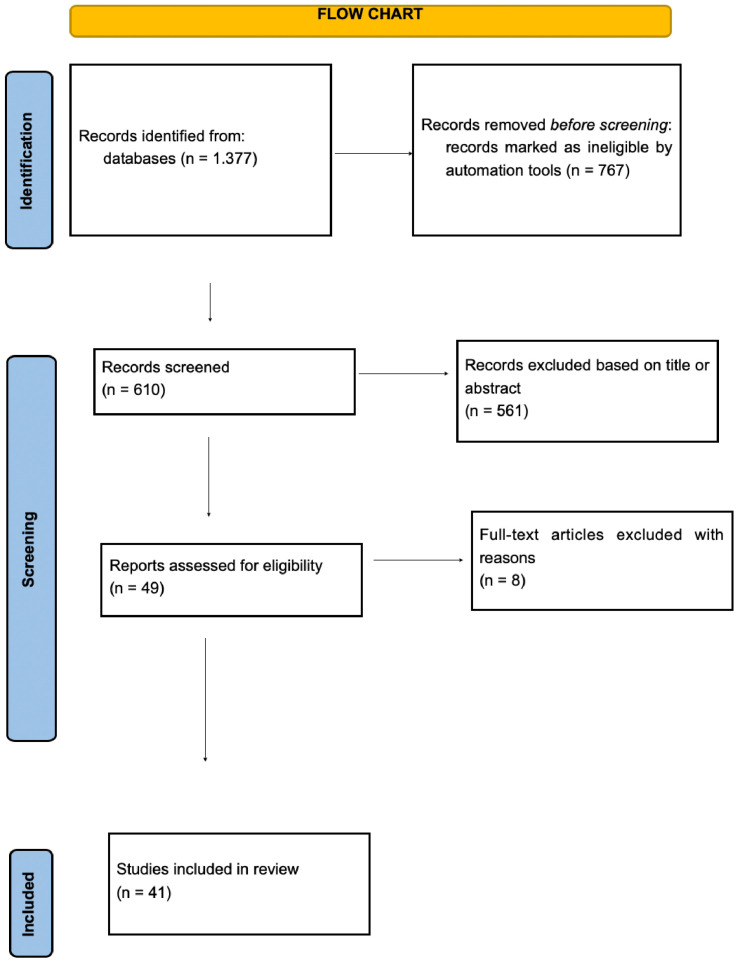
Study flowchart until final study selection.

**Table 1 biomedicines-12-02930-t001:** Studies reviewed and reported outcomes. TP: total pregnancy. SP: spontaneous pregnancy. MAR: medically assisted reproduction (number and percentage through MAR). NS: not specified. AMH: Anti-Müllerian hormone (in Nanograms/milliliter). AFC: Antral follicular count.

Author, Year	Study Design	Mean Age (Years)	Number of Patients	Control Group	Recurrence	TP	SP	MAR	AMH	AFC
Esinler et al., 2006 [12]	Retrospective	Unilateral 31.3 ± 3.9, bilateral 31.2 ± 4.4	57 (34 unilateral, 23 bilateral)	99 patients with tubal factor infertility	NS	NS	NS	Unilateral 45.2%, bilateral 44.4%	NS	NS
Yamamoto et al., 2009 [13]	Retrospective	<44 (details on text for each group)	287	Cyst-free with laparoscopic electrocoagulation of endometriosis lesions	NS	-	U:20/47 (43%), B: 7/38 (18%), u: 11/37 (30%), b: 5/22 (23%)	U:13/17 (77%), B: 9/24 (38%), u: 7/12 (58%), b: 8/11(73%)	NS	NS
Hsieh et al., 2009 [14]	Retrospective	Group 1: 33.1 ± 4.9, Group 2: 36.6 ± 6.5	108	-	Group 1: 25 (32.1%), Group 2: 4 (13.3%)	NS	NS	NS	NS	Group 1: 13.9 ± 6.1, Group 2: 11.9 ± 5.5
Hwu et al., 2011 [15]	Retrospective	33.27 ± 4.09	31	Infertile patients without endometrioma	NS	NS	NS	NS	2.01 ± 0.21	NS
Roman et al., 2011 [16]	Retrospective	31.6 ± 5.2	15	Cystectomy	NS	NS	NS	NS	NS	5.5 ± 3.9
Tang et al., 2013 [17]	Retrospective	<4 cm: 30.4 ± 3.9, >4 cm: 28.7 ± 4.0	85	-	<4 cm: 9 (25.7%), >4 cm: 13 (26%)	NS	NS	<4 cm: 19 (37.3%), >4 cm: 24 (38.1%)	NS	<4 cm: 4.9 ± 2.7, >4 cm: 4.9 ± 3.2
Lee et al., 2014 [18]	Retrospective	Cystectomy: 33.6 ± 2.9, AEST: 34.6 ± 4.9	101	No surgical intervention	NS	NS	NS	Cystectomy: 13/36 (36.1%), Sclerotherapy: 12/29 (41.3%)	NS	NS
Harada et al., 2015 [19]	Retrospective	37.0 ± 3.4	21	Ovaries without endometrioma excision	NS	NS	NS	40%	NS	NS
Lockyer et al., 2019 [20]	Retrospective	31.8 ± 5.9	21	-	2 (9.5%)	6 (46.2%)	0	6 (100%)	NS	NS
Chen et al., 2021 [21]	Retrospective	28.65 ± 3.66	46	Drainage and bipolar coagulation of OMA cyst wall	0	27 (71.05%)	16 (59%)	11 (41%)	3.40 ± 1.35	NS
Miquel et al., 2020 [22]	Retrospective	31.5 ± 4.5	74	No-EST group	NS	NS	NS	29 (43.3%)	3.2 ± 2.6	11.1 ± 6.3
Crestani et al., 2023 [23]	Retrospective	33.2 ± 4.7	69	-	6 (11.8%)	18 (40.1%)	7 (39%)	11 (61%)	2 ± 1.7	12 ± 9.6
Alborzi et al., 2004 [24]	Prospective	Group 1: 28.4 ± 5.8, Group 2: 28.5 ± 5.5	100	-	Group 1: 9/52 (17.3%), Group 2: 15/48 (31.3%)	NS	Group 1: 59.4%, Group 2: 23.3%	NS	NS	NS
Tsolakidis et al., 2010 [25]	Prospective randomized	29.9 ± 1.8	10	Cystectomy	2 (20%)	NS	NS	NS	3.99 ± 0.6	4.36 ± 0.8
Var et al., 2011 [26]	Prospective	27.04 ± 3.90	48	Coagulation with bipolar current	0	NS	NS	NS	NS	3.67 ± 1.26
André et al., 2011 [27]	Prospective	33.86 ± 3.56	21	-	NS	NS	NS	3/15 (20%)	NS	NS
Celik et al., 2012 [28]	Prospective	28.4 ± 5.7	65	-	NS	10 (15.4%)	NS	NS	0.72 ± 0.79	6.4 ± 2.2
Urman et al., 2013 [29]	Prospective	32.7 ± 6.1	25	-	0	3 (43%)	2/3 (67%)	1/3 (33%)	NS	NS
Uncu et al., 2013 [30]	Prospective	29.0 ± 5.4	30	No ovarian cyst	NS	NS	NS	NS	2.8 ± 2.2	9.7 ± 4.8
Aflatoonian et al., 2013 [31]	Prospective	29.4 ± 5.76	40	Patients with endometrioma for IVF without sclerotherapy	NS	NS	NS	6 (33.3%)	NS	NS
Kwon et al., 2014 [32]	Prospective	31.72 ± 5.71	68 (42 unilateral, 26 bilateral)	Non-endometriotic ovarian cyst	NS	NS	NS	NS	3.22 ± 2.09	NS
Bhat et al., 2014 [33]	Prospective	29.2 ± 3.6	73	-	NS	NS	NS	NS	NS	4.1 ± 1.5
Roman et al., 2014 [34]	Prospective	30.6 ± 4.8	22	-	1 (5%)	8 (73%)	3 (37%)	5 (63%)	3.1 ± 2.2	NS
Roman et al., 2015 [35]	Prospective	31.4 ± 5.1	124	-	18 (14.5%)	51 (61.4%)	33 (64.7%)	18 (35.3%)	NS	NS
Chun-Hua Zhang et al., 2015 [36]	Prospective	31.8 ± 8.2	207	-	0	NS	NS	NS	Group A: 2.0 ± 0.9 Group B: 2.0 ± 1.0 Group C: 3.1 ± 1.6	Group A: 4.2 ± 1.5 Group B: 4.0 ± 1.2 Group C: 6.3 ± 2.0
Muzii et al., 2015 [37]	Prospective	33.9 ± 2.3	28	Primary surgery group	0	NS	NS	NS	NS	5.1 ± 2.8
Muzii et al., 2016 [38]	Prospective Randomized	32.9 ± 5.7	51	Cystectomy	1 (2%)	NS	NS	NS	NS	4.4 ± 2.3
Kovačević et al., 2018 [39]	Prospective	30.3 ± 4.5	54 (37 unilateral, 17 bilateral)	-	NS	24 (77.4%)	18 (75%)	6 (25%)	1.72 ± 1.23 (unil.), 0.89 ± 0.82 (bil.)	NS
Bourdon et al., 2018, [40]	Prospective	33.7 ± 4.0	201	Women without OMA undergoing ART procedure	NS	NS	NS	53/151 (35%)	NS	NS
Saito et al., 2018 [41]	Prospective	37 bilateral, 32 unilateral	34 (10 bilateral, 24 unilateral)	Vaporization with bipolar current (unilateral and bilateral)	0	NS	NS	NS	2.5 ± 1.7 (unilateral)/0.8 ± 0.7 (bilateral)	NS
Candiani et al., 2018 [8]	Prospective randomized	32.1 ± 4.8	30	Cystectomy	0	3 (25%)	3 (100%)	NS	1.9 ± 0.9	8.6 ± 4.2
Yu Liang et al., 2019 [42]	Prospective	33.85 ± 5.47	41	Non-surgery	NS	NS	NS	46.2%	1.96 ± 1.00	8.69 ± 4.84
Shaltout et al., 2019, [43]	Prospective	20–35	200	-	Group D 27.1% Group C 24.4% Group DS 10.9% Group CS 9.1%	NS	10 with surgicel, 7 without surgicel (*p* = 0.543)	NS	Decrease in all groups with *p* < 0.001	Higher decrease in Group D
Hosseinimousa et al., 2020 [44]	Prospective	30.3 ± 4.61	79	GnRH-a alone	NS	NS	NS	12 (36.36%)	NS	NS
Kitajima et al., 2020 [45]	Prospective	31.4 ± 4.7	23	-	NS	3	2	1	Significant decline in one-step surgery	NS
Rius et al., 2020 [46]	Prospective randomized	32.13 ± 6.56	16	Cystectomy	0	3 (33.3%)	NS	NS	NS	9.33 ± 6.2
Alborzi et al., 2021 [47]	Prospective	Surgery: 31.09 ± 3.95, EST: 30.8 ± 4.69	101	-	Surgery: 14%, EST:34.1%	NS	NS	Surgery: 24/57 (42.1%), EST: 15/44 (34.1%)	NS	NS
Anh et al., 2022 [48]	Prospective	28.5	104 (77 unilateral, 27 bilateral)	-	NS	NS	19	NS	2.39 (1.44–3.87) unil., 0.92 (0.32–1.23) bil.	NS
Fakehi et al., 2022 [49]	Prospective	29.08 ± 4.6	100	-	NS	NS	NS	NS	Decline rate 30.07 ± 2.30%	NS
Ghasemi Tehrani et al., 2022 [50]	Prospective	31.46 ± 4.71	70	-	Sclerotherapy: 17 (48.57%), Laparoscopy: 15 (42.85%)	NS	NS	NS	Laparoscopy: 1.62 ± 1.22, Sclerotherapy: 2.09 ± 1.01	NS

## Data Availability

Available from A.I. or N.M. upon reasonable request.

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
