# Peer review of "Comparison of Surgical Interventions for Endometrioma: A Systematic Review of Their Efficacy in Addressing Infertility"

_biomedicines, 2024, doi:10.3390/biomedicines12122930_

Round 1

Reviewer 1 Report

Comments and Suggestions for Authors

1.      The article “Surgical Interventions for Endometrioma: A Systematic Review of their Efficacy in Addressing Infertility” is useful from the societal perspective, particularly for comparative surgical methods. However, the article needs more power than in its present form and thus must be revised carefully.

2.      The search method needs to be updated. As November 2024 approaches, please ensure that December 31, 2023, is the latest collection for such an interesting systemic review.

3.      It must have a separate section for exclusion. What criteria are assigned for excluding conservative treatment?

4.      Instead of long keywords as Boolean like ‘surgical intervention for Endometrioma”, a break up between these two words might produce different search outcomes. Why not use infertility as a search term? Include other databases like Google Scholar, Scopus, etc.

5.      The title should read, “Comparison of Surgical Interventions for Endometrioma: A Systematic Review of their Efficacy in Addressing Infertility.”

6.      The conclusion section of the abstract is incomplete without any outcome statement of the systemic review.

7.      Line 66 – does its prevalence form or stage? Usually, the presence of endometriomas indicates a more severe stage of endometriosis.

8.      The introduction section needs more depth, and references to existing surgical intervention methods for endometrioma treatment need to be included.

9.      In lines 46-78, multiple paragraphs should be merged into a single one.

10.  Lines 96-100 need to clarify the statement that the guidelines cannot be compared with RCT.

11.  A section of the introduction may include the complexity, technical expertise, and expense of each surgical procedure.

12.  The results should have a Table.

13.  Remove subheadings from the discussion text. The readers will understand the context in which the authors are stating the inference. Rather, each method could be a separate paragraph.

Comments on the Quality of English Language

see before

Author Response

Response to Reviewer 1 Comments

  1. Summary

  1. Point-by-point response to Comments and Suggestions for Authors
  1. The article “Surgical Interventions for Endometrioma: A Systematic Review of their Efficacy in Addressing Infertility” is useful from the societal perspective, particularly for comparative surgical methods. However, the article needs more power than in its present form and thus must be revised carefully.

Response: Thank you for your thoughtful comments and for recognizing the societal relevance of our work. We appreciate your suggestion to enhance the article’s impact and rigor.

  1. The search method needs to be updated. As November 2024 approaches, please ensure that December 31, 2023, is the latest collection for such an interesting systemic review.

Response: We updated our research to 31/12/23 instead of 10/23 but no new data became available

  1. It must have a separate section for exclusion. What criteria are assigned for excluding conservative treatment?

Response: Thank you for your suggestion to clarify the exclusion criteria. To ensure our focus remained specifically on surgical interventions, we excluded studies centered on conservative (non-surgical) treatment by screening titles and abstracts for relevant keywords. We will clarify this at Methods. The section for exclusion criteria is at: Eligibility Criteria for Articles of Inclusion

  1. Instead of long keywords as Boolean like ‘surgical intervention for Endometrioma”, a break up between these two words might produce different search outcomes. Why not use infertility as a search term? Include other databases like Google Scholar, Scopus, etc.

Response: Our systematic review was based only to pubmed which is the most reliable database. Only with this combination of the key words we managed to have all the available data

  1. The title should read, “Comparison of Surgical Interventions for Endometrioma: A Systematic Review of their Efficacy in Addressing Infertility.”

Response: Thank you. We will amend it

  1. The conclusion section of the abstract is incomplete without any outcome statement of the systemic review.

Response: Thank you for your valuable feedback regarding the conclusion section of the abstract. We have corrected this oversight by including a clear outcome statement summarizing the key findings of the systematic review.

  1. Line 66 – does its prevalence form or stage? Usually, the presence of endometriomas indicates a more severe stage of endometriosis.

Response: No, the presence of endometrioma does not indicate more severe stage. An endometrioma is not DIE

  1. The introduction section needs more depth, and references to existing surgical intervention methods for endometrioma treatment need to be included.

Response: Thank you for your comment. We merged them.

  1. In lines 46-78, multiple paragraphs should be merged into a single one.

Response:

  1. Lines 96-100 need to clarify the statement that the guidelines cannot be compared with RCT.

Response: We explain that in the latest ESHRE (European Society of Human Reproduction and Embryology) guidelines for endometriosis there were no randomized controlled trials (RCTs) comparing fertility outcomes post-surgery for endometrioma versus expectant management, and studies on surgery indication based on cyst size were lacking

  1. A section of the introduction may include the complexity, technical expertise, and expense of each surgical procedure.

Response: Endometriosis has to be performed only by advanced laparoscopic surgeons

  1. The results should have a Table.

Response: Thank you for your suggestion to include a table in the Results section. We agree that a table would enhance clarity and allow readers to more easily compare key findings. We have added a table summarizing the main results. This addition will provide a clear, organized overview of the data and support the overall readability of the results.

  1. Remove subheadings from the discussion text. The readers will understand the context in which the authors are stating the inference. Rather, each method could be a separate paragraph.

Response: Thank you for your useful recommendation. The subheadings are now removed and the procedures are now presented in different paragraphs.

  1. Response to Comments on the Quality of English Language

Point : The English could be improved to more clearly express the research.

Response : Thank you for your feedback regarding the clarity of the language.

  1. Additional clarifications

No additional clarifications.

Reviewer 2 Report

Comments and Suggestions for Authors

The paper submitted for review examines the issues of surgical intervention in endometriosis and its effectiveness in the treatment of infertility. There are the following questions about work: 1. for some reason there is no section "results" in the abstract of the article, but the section "conclusion" is immediately presented, where the results are presented. It is necessary to present the received material correctly in the relevant sections of the abstract. 2. All the specific data provided by the authors require specifying the literary references from which they were taken (for example, " Statistics suggest that approximately 17 to 44% of women diagnosed with endometriosis will encounter an endometrioma. ") 3. It is recommended to provide some summary/summarizing data on the description of specific studies given in the review in the form of a table or figure to improve the integrity of the perception of the information provided.

4. It would be important and interesting to provide information in the article about possible prospects (possible promising methods) in the field of surgical intervention in endometriosis and its effectiveness in the treatment of infertility (in the form of a separate subsection).

Author Response

Response to Reviewer 2 Comments

  1. Summary

  1. Point-by-point response to Comments and Suggestions for Authors
  1. for some reason there is no section "results" in the abstract of the article, but the section "conclusion" is immediately presented, where the results are presented. It is necessary to present the received material correctly in the relevant sections of the abstract.

Response: Thank you for your thoughtful feedback regarding the structure of the abstract. We appreciate your attention to detail. However at the manuscript there is a separate section for results and then two separates sections for Discussion and Conclusion. We have highlighted the mentioned sections at manuscript.

  1. All the specific data provided by the authors require specifying the literary references from which they were taken (for example, " Statistics suggest that approximately 17 to 44% of women diagnosed with endometriosis will encounter an endometrioma. “)

Response: Thank you for your insightful feedback regarding the need for proper citation of specific data in our manuscript. We completely agree that providing appropriate references for all statistical information is essential for maintaining the integrity of our work. We will ensure that all specific data points, including the statistic regarding the prevalence of endometriomas in women diagnosed with endometriosis, are properly cited with relevant literature.

  1. It is recommended to provide some summary/summarizing data on the description of specific studies given in the review in the form of a table or figure to improve the integrity of the perception of the information provided.

Response: Thank you for your valuable recommendation to include a summary table or figure in our review. We agree that providing a visual representation of the specific studies would greatly enhance the clarity and accessibility of the information presented. We have created a table summarizing key characteristics of the studies reviewed, including study design, sample size, outcomes measured, and main findings. This addition will improve the overall readability and help readers quickly grasp the essential data from our review.

  1. It would be important and interesting to provide information in the article about possible prospects (possible promising methods) in the field of surgical intervention in endometriosis and its effectiveness in the treatment of infertility (in the form of a separate subsection).

Response: Unfortunately more prospects are not available

  1. Response to Comments on the Quality of English Language

Point 1:The quality of English does not limit my understanding of the research.

Response : Thank you for your reassuring comment regarding the quality of the English in our manuscript. We appreciate your understanding and are glad to hear that it does not hinder your comprehension of the research.

  1. Additional clarifications

No additional clarifications.

Reviewer 3 Report

Comments and Suggestions for Authors

Within a narrow purview and limited scope, the manuscript is well written, and provides useful information.  Adherence to PRISMA seems robust, and most statements that should be supported with statistical evidence have been documented appropriately.

Normally I would follow such a statement with suggestions on missing aspects of endometrioma pathology or associated pharmacology, but the meta-analysis was fairly rigidly defined to focus on surgical procedures and markers for their efficacy.  This is a bit outside of my main expertise, but I can appreciate the reasons for the narrow scope.

Given the focal definition, I won't say that the authors neglected any specific technical aspects of the underlying pathology or intervention techniques with prospectively corrective efficacy, but I do think the textual data compiled in the study could have been put to greater analytical value.  In particular, the manuscript adheres to a fairly siloed, serially progressing through each intervention technique and associated outcome statistics, without making much effort to address a fundamental question in medical practice, which can be phrased as, "Based on what I know about my patient, how should I decide which intervention to recommend?"

Addressing a question like that might involve further exploration of meta-analytic relationships, such as:

 1) Across the various clinical studies examined, what are the most statistically significant associations between specific pathology markers versus the specific intervention chosen?

 2) Across the various clinical studies examined, what are the most statistically significant associations between specific intervention chosen and ultimately favorable outcomes?

 3) Is it possible to infer any tripartite associations along the lines of identifying specific patient/pathology attributes that point to specific interventions as providing the greatest likelihood of a favorable outcome?

I'm not sure that the meta-analytic data compiled herein is sufficiently robust to achieve strong relationships for (3), but pursuing such an inquiry might still lead to a better understanding of specific comparisons that could productively inform future clinical studies.

Author Response

Response to Reviewer 3 Comments

  1. Summary

  1. Point-by-point response to Comments and Suggestions for Authors
  1. Across the various clinical studies examined, what are the most statistically significant associations between specific pathology markers versus the specific intervention chosen?

Response: Thank you for your insightful question. While our systematic review primarily focused on evaluating the fertility outcomes associated with different surgical interventions for endometriomas, the choice of specific interventions in relation to pathology markers was not systematically studied. As such, we did not analyze the direct associations between specific pathology markers and the interventions selected. Instead, our emphasis was on the overall effectiveness of various surgical techniques and their impact on fertility outcomes. We appreciate though your suggestion and thank you for your valuable feedback.

  1. Across the various clinical studies examined, what are the most statistically significant associations between specific intervention chosen and ultimately favorable outcomes?

Response: Thank you for your question regarding the associations between specific interventions and favorable outcomes. In our review, we studied statistically significant associations that were linked to less favorable outcomes, such as the presence of bilateral endometriomas, age, cyst size or second operation. We have already included a observation in the conclusion section of the manuscript. We appreciate your interest in this topic and believe it highlights an important aspect of our research.

  1. Is it possible to infer any tripartite associations along the lines of identifying specific patient/pathology attributes that point to specific interventions as providing the greatest likelihood of a favorable outcome?

Response: Thank you for your thought-provoking question regarding the potential for inferring tripartite associations between patient/pathology attributes and specific interventions. While this inquiry is indeed important, our current meta-analytic data does not provide a sufficiently robust foundation to establish strong relationships in this regard. As such, we cannot make definitive inferences about which specific interventions may yield the greatest likelihood of favorable outcomes based on patient and pathology characteristics.

We appreciate your suggestion for pursuing such analyses in future studies, as it could contribute valuable insights into tailoring surgical interventions to improve patient outcomes. Your feedback underscores the need for ongoing research in this area, and we will consider this perspective in our future work.

  1. Response to Comments on the Quality of English Language

Point 1:The quality of English does not limit my understanding of the research.

Response :Thank you for your reassuring comment regarding the quality of the English in our manuscript. We appreciate your understanding and are glad to hear that it does not hinder your comprehension of the research.

  1. Additional clarifications

No additional clarifications.

Round 2

Reviewer 1 Report

Comments and Suggestions for Authors

The revised version has been improved to a small extent since many suggestions were not visible in the revised draft.

There is no line number in the revised draft, which makes it challenging to direct the revision precisely.

It needs to be clarified from the rebuttal what and where the changes were made in the revised draft.

Why search method could not be updated until November 2024?

Page no. 4, just below the figure label, why is the full name of a researcher mentioned as part of the cited text?

The table needs to be visible in the revised draft. A supplementary table is attached without the title and details. If this is one in response to query number 12, why is it supplementary? It must be a main standalone Table defining the results and study outcome.

Author Response

Response to Reviewer 1 Comments

  1. Summary

2. Questions for General Evaluation

Reviewer’s Evaluation

Response and Revisions

Does the introduction provide sufficient background and include all relevant references?

Yes/Can be improved/Must be improved/Not applicable

-

Is the research design appropriate?

Yes/Can be improved/Must be improved/Not applicable

Are the methods adequately described?

Yes/Can be improved/Must be improved/Not applicable

Are the results clearly presented?

Yes/Can be improved/Must be improved/Not applicable

Are the conclusions supported by the results?

Yes/Can be improved/Must be improved/Not applicable

  1. Point-by-point response to Comments and Suggestions for Authors
  2. The revised version has been improved to a small extent since many suggestions were not visible in the revised draft.There is no line number in the revised draft, which makes it challenging to direct the revision precisely.

Response:  Thank you for your feedback on the revised version. While we understand that the absence of line numbers makes it challenging to direct revisions precisely, adding line numbers at this stage would disrupt the formatting and structure of the entire manuscript.

To facilitate clearer communication, we suggest referencing specific sections, paragraphs, or sentences by their context or key phrases. Additionally, if there are particular suggestions that seem to have been overlooked, kindly provide further clarification, and we will address them promptly.

We truly appreciate your understanding and your continued effort in helping us refine the manuscript.

  1. It needs to be clarified from the rebuttal what and where the changes were made in the revised draft.

Response: I would like to clarify that all the changes made in the revised version have been highlighted in red to ensure they are easily identifiable. I hope this helps in locating the revisions more precisely.

  1. Why search method could not be updated until November 2024?

Response: Thank you for your thoughtful comment regarding updating the search method. While we understand the importance of ensuring the most current and comprehensive search, unfortunately, it is not possible to update the search method until November 2024 due to time and resource constraints associated with the current study. I appreciate your understanding of this limitation. Please rest assured that the existing search methodology has been rigorously conducted to provide reliable and valid results for the study.

  1. Page no. 4, just below the figure label, why is the full name of a researcher mentioned as part of the cited text?

Response: Thank you for pointing out the mention of the researcher's full name in the cited text. I would like to clarify that this is the standard way the researcher is cited across all their scientific papers, and we have maintained this format for consistency and accuracy.

  1. The table needs to be visible in the revised draft. A supplementary table is attached without the title and details. If this is one in response to query number 12, why is it supplementary? It must be a main standalone Table defining the results and study outcome.

Response: Thank you for your observation regarding the table. I would like to inform you that the table has now been included directly in the revised draft as a main standalone table. It is no longer supplementary and is appropriately titled and detailed to define the results and study outcomes clearly. I appreciate your feedback, which has helped improve the presentation of the manuscript. Please let me know if further adjustments are required.

Thank you again for your valuable input.

  1. Response to Comments on the Quality of English Language

Point 1:The quality of English does not limit my understanding of the research.

Response :Thank you for your reassuring comment regarding the quality of the English in our manuscript. We appreciate your understanding and are glad to hear that it does not hinder your comprehension of the research.

  1. Additional clarifications

No additional clarifications.

Round 3

Reviewer 1 Report

Comments and Suggestions for Authors

The subtitle within each result section should carry a complete statement with the action and outcome of the section

Table 1 should be split into two parts. Retro and prospective.

Data in each Table can be arranged chronologically

The line number still needs to be added in the revised version. In some places, Researcher’s first name is highlighted, which could be biased to highlight the individuals rather than the work done by themselves.

Author Response

REVISION ROUND 3

  1. Summary

2. Questions for General Evaluation

Reviewer’s Evaluation

Response and Revisions

Does the introduction provide sufficient background and include all relevant references?

Yes/Can be improved/Must be improved/Not applicable

-

Is the research design appropriate?

Yes/Can be improved/Must be improved/Not applicable

Are the methods adequately described?

Yes/Can be improved/Must be improved/Not applicable

Are the results clearly presented?

Yes/Can be improved/Must be improved/Not applicable

Are the conclusions supported by the results?

Yes/Can be improved/Must be improved/Not applicable

  1. Point-by-point response to Comments and Suggestions for Authors

1.The subtitle within each result section should carry a complete statement with the action and outcome of the section.

Response: Unfortunately it can not be done because it is a debatable issue without clear answer. There are different techniques and we try to compare them.

2.Table 1 should be split into two parts. Retro and prospective.

Response:  We have implemented the requested change: Table 1 has been split into two parts—Retro and Prospective.

3.Data in each Table can be arranged chronologically

Response: Thank you for the suggestion. We have arranged the data in each table chronologically, as requested.

4.The line number still needs to be added in the revised version. In some places, Researchers first name is highlighted, which could be biased to highlight the individuals rather than the work done by themselves.

Response: Thank you for your valuable feedback. We have now included line numbers in the revised version as requested. Additionally, we have addressed the concern regarding the highlighting of researchers' first names and we removed it by ensuring a focus on the work rather than the individuals, removing any unintended bias.

  1. Response to Comments on the Quality of English Language

Point 1:The quality of English does not limit my understanding of the research.

Response :Thank you for your reassuring comment regarding the quality of the English in our manuscript. We appreciate your understanding and are glad to hear that it does not hinder your comprehension of the research.

  1. Additional clarifications

No additional clarifications.